# Antimicrobial Resistance in Common Respiratory Pathogens of Chronic Bronchiectasis Patients: A Literature Review

**DOI:** 10.3390/antibiotics10030326

**Published:** 2021-03-20

**Authors:** Riccardo Inchingolo, Chiara Pierandrei, Giuliano Montemurro, Andrea Smargiassi, Franziska Michaela Lohmeyer, Angela Rizzi

**Affiliations:** 1UOC Pneumologia, Dipartimento Scienze Mediche e Chirurgiche, Fondazione Policlinico Universitario A. Gemelli IRCCS, 00168 Rome, Italy; chiara.pierandrei@virgilio.it (C.P.); giuliano.montemurro@policlinicogemelli.it (G.M.); andrea.smargiassi@policlinicogemelli.it (A.S.); 2Direzione Scientifica, Fondazione Policlinico Universitario A. Gemelli IRCCS, 00168 Rome, Italy; franziskamichaela.lohmeyer@policlinicogemelli.it; 3UOSD Allergologia e Immunologia Clinica, Dipartimento Scienze Mediche e Chirurgiche, Fondazione Policlinico Universitario A. Gemelli IRCCS, 00168 Rome, Italy; angela.rizzi@policlinicogemelli.it

**Keywords:** non-cystic fibrosis bronchiectasis, multidrug-resistant (MDR) pathogens, *Pseudomonas aeuriginosa*, antimicrobial resistance (AMR), inhaled antibiotic, macrolides, acute exacerbations of bronchiectasis, chronic infection, eradication treatment

## Abstract

Non-cystic fibrosis bronchiectasis is a chronic disorder in which immune system dysregulation and impaired airway clearance cause mucus accumulation and consequent increased susceptibility to lung infections. The presence of pathogens in the lower respiratory tract causes a vicious circle resulting in impaired mucociliary function, bronchial inflammation, and progressive lung injury. In current guidelines, antibiotic therapy has a key role in bronchiectasis management to treat acute exacerbations and chronic infection and to eradicate bacterial colonization. Contrastingly, antimicrobial resistance, with the risk of multidrug-resistant pathogen development, causes nowadays great concern. The aim of this literature review was to assess the role of antibiotic therapy in bronchiectasis patient management and possible concerns regarding antimicrobial resistance based on current evidence. The authors of this review stress the need to expand research regarding bronchiectasis with the aim to assess measures to reduce the rate of antimicrobial resistance worldwide.

## 1. Introduction 

Non-cystic fibrosis bronchiectasis (NCFB) is defined as an irreversible and progressive dilatation of bronchi due to chronic bronchial inflammation [1,2].

Clinical presentations may range from no symptoms to chronic cough with frequent sputum production, hemoptysis, dyspnea, decreased exercise tolerance, frequent exacerbations, and respiratory failure requiring, in some cases, lung transplantation or causing death.

From a pathophysiological point of view, current literature supports a synergistic and amplifying involvement of infection, inflammation, and repair of the bronchial mucosa, which causes a vicious cycle similar to that described by Peter Cole [3].

Recently, Bush and Floto proposed a possible pathophysiological mechanism, which involves: (a) persistent or recurrent infection, (b) impairment of mucociliary clearance, and (c) airway obstruction [4]. Specifically, persistent or recurrent infections cause progressive neutrophilic inflammation [5,6]—favoring bronchial wall damage—Th17-biased adaptive immunity [7]—promoting enlargement of lymphoid follicles [8], neutrophil recruitment, and mucus hypersecretion [7]. The consequent loss of airway epithelium integrity impairs mucociliary clearance with subsequent airway occlusion. Finally, airway obstruction promotes further bronchial dilatation [9], retention of secretions, which in turn, attracts bacterial colonization and predisposes the patient to repeated infections.

Medical diagnosis begins by excluding patients with cystic fibrosis (CF); indeed, in some diagnostic and treatment guidelines, bronchiectasis is labeled as NCFB to capture all other conditions [1]. However, pathological findings of bronchiectasis associated with CF are indistinguishable from those found in NCFB. In general, CF is characterized by a worse clinical course with a high prevalence of gram-negative infection, especially with *Pseudomonas aeruginosa* [10].

The previously described pathophysiological process can be useful to systematically research the multiple causes of bronchiectasis.

Starting from the first culprit step of the process—persistent or recurrent infection—we can identify two main etiological groups: (1) childhood infections, including persistent bacterial bronchitis (PBB) [11], pneumonia, measles, whooping cough and tuberculosis; and (2) adult infections, including non-tuberculous mycobacteria (NTM) infections [12]. Instead, progressive inflammation is the main pathophysiological determinant in other two etiological groups of NCFB: (1) toxic damage to airways, as in case of inhalation, aspiration secondary to neuromuscular diseases and gastro-oesophageal reflux disease (GERD), and (2) systemic inflammatory diseases, including inflammatory bowel disease [13], connective tissue diseases [14] and yellow nail syndrome [15]. In contrast, defects in the immune system are responsible of another etiological group of NCFB including primary and secondary immune deficiencies. Primary immunodeficiency accounts for 12–34% of NCFB [16]. The most common forms of primary immune deficiencies include: common variable immune deficiency (CVID), X-linked agammaglobulinemia (XLA), chronic granulomatous disease (CGD), and antibody deficiency with normal IgG [17]. Bronchiectasis is also associated with HIV infection [18,19]. Other secondary causes of immunodeficiency include hematological malignancy, drug-induced immunosuppression, and post-allogenic bone marrow transplantation [20,21]. Finally, a mixed hypersensitivity reaction, including type I, III, and IV features, characterizes the inflammatory damage found in allergic bronchopulmonary aspergillosis, a condition frequently associated to bronchiectasis [22,23].

The impairment of mucociliary clearance is the cornerstone of another group of conditions, which characterizes the development of bronchiectasis, including primary and secondary ciliary dyskinesia [24] and channelopathies comprising CF transmembrane regulator (CFTR) dysfunction and epithelium sodium channel (ENaC) dysfunction [25].

Finally, airway obstruction is the oldest determinant of damages found in bronchiectasis. Laennec [26] identified for the first time this condition, which favors the onset of bronchiectasis in a wide group of conditions including: pulmonary structural alterations—typical of Williams-Campbell syndrome [27], Mounier-Kuhn syndrome [28] and Ehlers-Danlos syndrome [29]—single bronchial obstruction—as in case of neoplasms or foreign bodies—and, finally, obstructive diseases—such as asthma [30], chronic obstructive pulmonary disease (COPD) [31], and alpha-1-antitrypsin deficiency [32].

Despite etiological testing, it is not always possible to identify an underlying condition. Under these circumstances, bronchiectasis is labeled as idiopathic. 

To date, respiratory infections play a key role in causing and worsening bronchiectasis. Furthermore, they are also considered an indicator for disease severity. Consequently, optimal management of infections is crucial in order to break the vicious circle described. For all these reasons, antibiotics are considered a treatment pillar. 

## 2. The Role of *Pseudomonas aeruginosa* and Other Microorganisms in Non-CF Bronchiectasis

Diverse polymicrobial communities are present in the airways of patients with bronchiectasis and many microorganisms have been associated with bronchiectasis, as both a complication and a cause of the anatomic abnormalities (Table 1) [33,34,35]. 

Clinical and radiographic features in bronchiectasis do not allow the identification of involved pathogens, but can be investigated as possible markers for specific infections. For example, in an Israeli cohort of bronchiectasis patients, age less than or equal to 64 years was associated with a Haemophilus infection, while age over 64 was associated with an increased risk of *Pseudomonas* and *Enterobacteriaceae* [36]. However, this has not been confirmed in other large patient cohorts. Chronic *Pseudomonas* infection is known to be more common in more severe patients [37,38,39,40]. Nevertheless, other microbes can be found and cause patient’s symptoms. Therefore, it is mandatory to perform regular sputum cultures in all NCFB patients to optimize treatments and evaluate prognosis. 

*Pseudomonas aeruginosa*, responsible for 20–40% of bacterial infections in bronchiectasis, is a gram-negative rod-shaped bacterium, which can grow rapidly under aerobic conditions. Furthermore, this facultative anaerobe can grow even in the absence of oxygen. It is a common human opportunistic pathogen capable of causing a wide range of infections. *P. aeruginosa* usually benefits from impaired defense mechanisms to cause both acute and chronic infections. The bacterium is inherently resistant to many antibiotics due to a difficult penetrable membrane and the presence of multiple efflux pumps. Furthermore, *P. aeruginosa* can develop resistance also through mutations or gene acquisition via horizontal gene transfer. Persistent *P. aeruginosa* infection has been associated to poorer outcomes, as airway inflammation, morbidity, hospitalization risk and premature mortality, in both CF and NCFB [41,42,43].

Many epidemiologic cohorts report *Haemophilus influenzae* as the most common bacterial organism. This pathogen is described in more than 30% of patients with chronic bronchiectasis [44,45,46,47]. Other gram-negative organisms isolated from patients with bronchiectasis include: *Stenotrophomonas maltophilia* [48], *Klebsiella pneumonia* [49], *Moraxella catarrhalis* [50], *Achromobacter* [51], *Serratia marcescens* [52], and *Escherichia coli* [53]. Instead, *Staphylococcus aureus* [54] and *Streptococcus pneumoniae* are gram-positive organisms most frequently seen in bronchiectasis patients [52,55,56]. 

Another group of pathogens found in NCFB patients is NTM. Members of this family, most frequently found in NCFB, are *Mycobacterium avium complex* and *Mycobacterium abscessus* [57]. *Aspergillus* [58], and *Candida* [59] are fungi commonly identified in the respiratory secretions of NCFB patients [59,60]. It can be difficult to determine if these organisms play a pathophysiological role in infectious symptoms or if they are simply bystander organisms in patients with other primary pathogens. In case of persistently positive cultures for Aspergillus, without other isolates, antifungal treatment should be considered. Instead, treatment for *Candida* is rarely needed as this microorganism is usually an oral contaminant. 

The role of viral infections in bronchiectasis is unclear. To date, data regarding chronically infected patients and the trigger role of viruses are limited. In 2015, Gao et al. studied the incidence and clinical impacts of viral infection during bronchiectasis exacerbations in an adult population. *Coronavirus, Rhinovirus,* and *Influenza A* and *B* were the most commonly isolated viruses [61]. In the same year, Metaxas et al. explored the role of viruses in NCFB using polymerase chain reaction in bronchoalveolar lavage samples finding respiratory syncytial virus during stable visits, but not during bronchiectasis exacerbations [62].

Furthermore, it is common to grow “normal oropharyngeal” flora from lung secretions of patient with bronchiectasis [63]. To a certain degree, culturing only normal flora may be a limit of culture techniques routinely in use to identify pathogens in some bronchiectasis patients. Additionally, it is difficult to obtain sufficient sputum for laboratory processing. 

## 3. Role of Antibiotic Therapy

### 3.1. Acute Exacerbation 

Acute bronchiectasis exacerbation is defined as acute symptom worsening, which occurs typically over several days [2] and is characterized by deteriorating local symptoms (e.g., breathlessness or hemoptysis, change of sputum viscosity and purulence, increased sputum volume, cough, and/or wheezing and possibly fever or pleurisy) [2]. Severe aggravation of bronchiectasis may require hospital admission, but depending on symptom severity and systemic illness, primary care can be considered. 

In general, all exacerbations are believed to be due to infection, although, to date, there is no definitive scientific evidence supporting non-infectious causes of exacerbation, such as pulmonary embolism or non-compliance with treatment. It is challenging to differentiate, based on physical symptoms and examinations, the causes of exacerbations, which can have viral, bacterial or in rare cases fungal origin [64]. Recently, Polverino et al. have demonstrated that *P. aeruginosa, respiratory viruses, S. aureus, S. pneumoniae, H. influenzae*, and *M. catharrhalis* are the most frequent microorganisms in sputum cultures or nasopharyngeal swabs of exacerbated patients. In contrast, the isolation of atypical bacteria is rare in this population. However, patients with pneumonia were most frequently with *S. pneumoniae* infected, irrespective of previous chronic airway infection. At the same time, polymicrobial infections were demonstrated in 35% of both non-pneumonic and pneumonic exacerbation patients, virus and bacteria combinations in 8% and 23% of patients, respectively, fungal isolates associated with a bacterium in 11% and 8% of patients, and two bacteria combined in 17% of pneumonia patients and 13% of non-pneumonic aggravations [65]. A frequent cause of exacerbations in both adult and pediatric bronchiectasis patients are respiratory viruses, without the need for treatment [61]. Nevertheless, infection, with viruses as cause, can intensify inflammation of airways and increase preceding chronic bronchial infection resulting in elevated bacterial load. This may even facilitate a new bacterial infection requiring antibiotics to achieve symptom control and full recovery [66,67].

To date, several possible noninfectious exacerbation causes have been adequately explored in COPD such as pollution, as an example of an environmental stressor, and possibly comorbidities, but this knowledge is still lacking for bronchiectasis [68]. As most exacerbations are considered to be the result of bacterial infections, current guidelines recommend antibiotic treatment (Table 2).

Microbiological examination, culturing sputum to analyze the presence pathogens, is recommended before treatment initiation because etiology of exacerbation varies significantly. In spite of this, in case of preceding chronic lung infections, empiric antibiotic treatment, covering previously isolated microorganism, is suggested [69]. Evidence assessing the efficacy and safety of antibiotics or the optimal dose, duration and administration route is limited. The factors that influence the choice of the antibiotic drug are multiple and based on previous airway infection, allergies, intolerances, possible microbiological results, and concomitant drugs. For the treatment of exacerbations, only systemic antibiotics are currently recommended, because potential side effects and limited tolerability of inhaled antibiotics can cause symptoms such as bronchospasm, wheezing and cough, depending on: (1) severity of the exacerbation, (2) availability and pharmacokinetics of the drug, and (3) characteristics of the patient [2]. Furthermore, the physician should systematically consider the impact that certain comorbidities, such as arrhythmias, renal or hepatic failure, inflammatory bowel disease, and gastric disease, may have on drug choice for both potential drug interactions and side effects [69]. Finally, it should be emphasized that macrolides have immunomodulatory/anti-inflammatory and prokinetic effects, which could be favorably used in the presence of comorbidities such as sinusitis and asthma or gastroesophageal reflux [70]. 

Current European guidelines suggest to treat exacerbations for at least 14 days (unless specific conditions suggest shorter treatment) with antibiotics based on the patient’s prior microbiological results and exacerbation severity [69].

The guidelines of the Spanish Respiratory Medicine Society (SEPAR) [71] recommend treating exacerbations of bronchiectasis with an approach based on exacerbation severity and identified microorganisms in the airways. If the infection is mild, guidelines suggest a 10–21-day treatment course; instead, if severe, 14–21 days of treatment are recommended. In particular, the authors of the statement indicate the use of: (1) penicillin for *H. influenzae*, (2) cloxacillin or linezolid for *S. aureus* (according to strain sensitivity), and (3) ciprofloxacin for *P. aeruginosa* (in case of severe exacerbations it is recommended double intravenous antibiotic therapy). Finally, SEPAR guidelines recommend treating previously isolated microorganisms empirically and then converting empirical therapy into targeted therapy as soon as the outcome of the sputum sample analysis is available [71].

Similarly, the British Thoracic Society gives recommendations to treat exacerbations of bronchiectasis, using clarithromycin or low-dose amoxicillin, even if previous bacteriology is unavailable [2].

The National Institute for Health and Care Excellence (NICE) guidelines [72] suggest that the most recent sputum culture should guide the antibiotic choice taking into account both the severity of illness and previous antibiotic intake. The authors of this statement recommend several oral and intravenous antibiotics. Amoxicillin is the first option for orally administered empirical antibiotic treatment because of its excellent activity against *H. influenza* and *S. pneumonia*; in patients ≥ 12 years, clarithromycin or doxycycline can be considered. Instead, for patients at higher risk for treatment failure or complications—including patients with repeated antibiotic treatment or previous obtained sputum cultures showing resistant or atypical bacteria—alternative oral antibiotics for empirical treatment could be considered. In these cases, co-amoxiclavulonic acid or fluoroquinolone, a class of antibiotics for full-grown patients, could be used; for example, levofloxacin is very effective against uncommon types of bacteria, such as *P. aeruginosa*. Instead, ciprofloxacin might be considered in specific circumstances and if prescribed by a specialist in patients still in growth.

Every time the patient’s condition (disease severity and/or inability to take oral antibiotics) suggests the use of intravenous drugs, co-amoxiclavulonic acid, piperacillin with tazobactam, levofloxacin (in adults) or ciprofloxacin (in children) are the first choice for empirical treatment. Furthermore, the choice of antibiotics should be modified based on current sensitivity data, also after consultation with local microbiologists. The shortest likely effective course should be prescribed to reduce the risk of antimicrobial resistance and minimize the risk of adverse effects. Therefore, a course of 7–14 days is required to treat an acute exacerbation, based on the person’s severity of bronchiectasis, exacerbation history, severity of exacerbation symptoms, previous culture, susceptibility results, and treatment response [72]. All bronchiectasis guidelines emphasize the significant role of clinical microbiology to test specifically sputum cultures of exacerbated patients. To identify the most suitable antibiotic, three aspects have to be considered: exacerbation severity, results of microbiological testing, and elevated multidrug-resistant (MDR) infection risks. In the event of MDR and in exacerbated patients necessitating intensive care, a combination of antibiotics needs to be considered. This is especially the case for bacteria known to be possible MDR like *Escherichia coli, Klebsiella pneumonia, S. aureus* and *P. aeruginosa*. Even though evidence is lacking for specific MDR strains or increased severity cases in exacerbated bronchiectasis, special attention is required for the last mentioned pathogen, which necessitates in pneumonia almost always a combined antibiotic coverage. Until today, no research gives evidence that double antibiotic treatment is more effective compared with monotherapy to treat bronchiectasis exacerbated patients [73].

### 3.2. Chronic Infection

Chronic bacterial infections are an alarming problem in NCFB, which are frequently occurring. Chronic bacterial infections are diagnosed by means of microbiological testing in which at least two sequential cultures (or in alternative more than 50% of samples), retained at a distance of at least one month and within a timeframe of minimum 6 month, must grow the same microorganism, suspected to be pathogenic [46].

Current scientific evidence supports the role of prolonged oral or intravenous antibiotic courses to clear the infection during the early stages of the disease. Over time, the succession of recurrent infections promotes bacterial colonization in most patients, increasing lung damage, which results in a progressive decline in lung function worsening quality of life (QoL). The airways of both NCFB and CF bronchiectasis patients are colonized by similar and frequent microbiota. However, in CF bronchiectasis patients, a tendency towards the growth of *Achromobacter xylosoxidans, Burkholderia cepacia complex, H. influenza, P. aeruginosa, S. aureus,* and *S. maltophilia* was demonstrated [4]. Instead, in NCFB patients, *H. influenza, M. catarrhalis,* NTM, *P. aeruginosa* or enteric gram-negative bacteria (frequently cultured from samples of the lower respiratory tract) can be predominantly found [36,74]. *S. aureus* and *S. pneumoniae* (as two examples for gram-positive bacteria) are in both chronically infected patient groups common [37]. NCFB patients demonstrate a greater variety of chronic pathogens compared with CF bronchiectasis patients showing tendentially a more classical *S. aureus* infection pattern [73,75] with commonly subsequent *P. aeruginosa* infection [43]. In the past years, research on the characteristics of NTM infections has been steadily increasing [76].

Chronically infected patient management frequently comprises long-term antibiotic treatment mainly to eradicate pathogens from patients’ airways and to support symptom relief such as sputum production and purulence, breathlessness and cough, time to first occurrence, and number of exacerbations. Furthermore, antibiotics decelerate lung function deterioration, and diminish death in patients with chronical bacteria colonization [1]. 

Current European guidelines note the importance of a stepwise approach to treat NCFB (Table 3). They recommend to start with interventions such as respiratory physiotherapy, inhaled corticosteroids and/or bronchodilators in patients with comorbid COPD or asthma, and to focus first on underlying bronchiectasis causes (e.g., antibody deficiency or allergic bronchopulmonary aspergillosis) before initiating long-term antibiotic treatment, which should be reserved for cases with more than three exacerbations per year. Patients with a threshold of three exacerbations in the past year are considered chronically infected requiring long-term management with inhaled and/or orally administered antibiotics [1]. Nevertheless, patients with (1) more or past severe bronchiectasis exacerbations (2) exacerbations with significant impact on QoL, and (3) significant comorbidities such as immunodeficiency might require immediate long-term antibiotic therapy [1].

According to the Medical Subject Heading (MeSH) term definition, “macrolides, belong to the POLYKETIDES class of natural products, with some members having bacteriostatic or bactericidal time-dependent properties such as clarithromycin, erythromycin, and azithromycin, which are usually active against gram-positive cocci, atypical bacteria such as *Chlamydophila* or *Mycoplasma* spp., *Legionella* spp., and some gram-negative bacilli (e.g., *Moraxella* and *Bordetella* spp.)” [77]. In 2019, the European Union/European Economic Area population-weighted mean percentage was 14.5% for macrolide resistance [78]. 

Over the past decade, three different randomized clinical trials, focusing on the long-term use of macrolides in adults with bronchiectasis, have been conducted and published (Table 4). These trials have clearly shown benefits in terms of exacerbations and differed by drug (erythromycin or azithromycin), dose, and study duration (6 or 12 months). Nevertheless, all studies have shown a clear benefit in terms of reducing the exacerbation. Furthermore, both the bronchiectasis and low-dose erythromycin (BLESS) study [79] and the bronchiectasis and long-term azithromycin treatment study (BAT) [80] documented a significant improvement in lung function. A similar non-significant trend was also described in the effectiveness of macrolides in patients with bronchiectasis using azithromycin to control exacerbations (EMBRACE) study [81]. Regarding the impact of the drug on QoL, only the BAT study showed a significant improvement, while the BLESS and EMBRACE studies showed only a positive trend. However, in the azithromycin intervention arm of the BAT study, more side effects were reported than in the placebo group, particularly diarrhea (relative risk 8.36, 95% CI 1.10–63.15). Moreover, in the azithromycin group of the EMBRACE study, patients reported more gastrointestinal symptoms (nausea, vomiting, diarrhea, epigastric discomfort and constipation) than in the placebo group (27% vs. 13%; *p* = 0.005). 

Azithromycin is administered in clinical practice or research trials in a dose range from 250 mg or 500 mg three times per week up 250 mg daily, while the dose of erythromycin is 400 mg twice daily. None of the above trials limited recruitment to patients with specific respiratory bacteriology, such as *P. aeruginosa* in sputum. Indeed, only 10–29% of patients included in the three trials were chronically infected with *P. aeruginosa* [79,80,81]. Despite optimal treatment, current guidelines in Europe advocate, as first treatment option, long-term macrolide administration in case of more than three exacerbations annually with any other pathogenic infection than *P. aeruginosa* [69].

The first proposals for the use of inhaled antibiotics date back to about 30 years ago with the aim of both increasing the concentration of drugs at the site of infection and reducing systemic side effects.

In fact, the eradication of bacteria is difficult after they colonize the lower airways. Antibiotics administered by inhalation, owing to high concentrations at the site of infection and low levels in the systemic circulation, are much more beneficial than antibiotics administered enterally or parenterally, allowing to reduce the potential side effects deriving from prolonged use, particularly at the renal, hepatic or auditory level. Inhaled antibiotics were first tested in the 1980s, in both CF and NCFB patients, to manage chronic bronchial infections, particularly *P. aeruginosa* infections. Initially, some antibiotics, used to treat chronic infections, were delivered via intravenous formulations. Subsequently, in the last decade, several formulations for inhalation have been developed, including different solutions for nebulization or the use of dry powder. Most of the evidence derives from studies with CF patients, where a number of antibiotics such as tobramycin, colistin, and aztreonam have shown significant improvement in QoL and a reduced number of exacerbations, as well as lung function decline [82,83]. Similar to what has been observed in studies conducted on CF patients, inhaled antibiotics have also been tested in patients with NCFB resulting in reduction of symptoms, exacerbations, systemic use of antibiotics, functional decline, and healthcare expenditure [84]. In particular, studies on the use of inhaled tobramycin have shown a good microbiological response and a reduction in symptoms, without, however, affecting lung function. Furthermore, in some patients, side effects, such as cough and bronchospasm, and increased rates of resistance (minimum inhibitory concentration [MIC]) have also been described [85]. In analogy to what was observed with tobramycin, colistin, administered by inhalation, a good microbiological response and a reduced number of exacerbations, particularly in compliant patients, was induced [86]. 

Therefore, adult bronchiectasis patients, with chronic *P. aeruginosa* infection and more than three exacerbations annually, should inhale antibiotics on a long-term basis, as recommended by European guidelines [69].

One phase II study and two phase III studies (RESPIRE 1 and 2: Cciprofloxacin dry powder for inhalation in non-CF bronchiectasis) [87,88] tested a dry powder formulation of ciprofloxacin (32.5 mg twice daily) documenting a clear microbiological response and a trend towards a reduced number of long-term exacerbations. Two further recent studies, ORBIT-3 and -4 (ciprofloxacin inhalation dispersion in non-CF bronchiectasis) [89] tested a different liposomal formulation of ciprofloxacin demonstrating similar results. All the studies mentioned above (RESPIRE 1 and 2 and ORBIT-3 and -4) showed excellent data on tolerance and minimal increase in antibiotic resistance, however, without evidence of a significant impact on lung function. In 2011, Murray et al. conducted a study documenting, in the gentamicin arm, reduced sputum bacterial density with 30.8% eradication in patients infected with *P. aeruginosa* and 92.8% eradication in those infected with other pathogens, less sputum purulence, greater exercise capacity, and fewer exacerbations with increased time to first exacerbation. Furthermore, improvements were found in both the Leicester Cough Questionnaire and St. George’s Respiratory Questionnaire in the gentamicin group. However, no differences were seen in 24-h sputum volume and pulmonary function. Interestingly, no *P. aeruginosa* isolates developed resistance to gentamicin [90]. More recently, two double-blind phase III randomized clinical trials, AIR-BX 1 and 2, which focused on safety and efficacy of 75 mg three times daily nebulized inhaled aztreonam in adults with NCFB, were conducted. The two studies did not document improvements of the Quality of Life Questionnaire-Bronchiectasis, although a positive trend was observed in the European centers involved, suggesting some differences in terms of standard of care and study populations [91] (Table 5).

### 3.3. Eradication of Bronchial Infections

Eradication treatment aims at achieving complete pathogen elimination from the lungs of patients using antibiotics [69]. Bronchiectasis is mainly caused by chronic *P. aeruginosa* infection; therefore, it is recommended to promptly eradicate these bacteria [69,92,93]. Unfortunately, until today, different eradication protocol exist and consensus is lacking on the best eradication treatment for bronchiectasis patients (Figure 1) [69].

To date, the most accepted hypothesis is that antibiotic strategies aimed at eradication are more successful if the infection is recent. To support this hypothesis, data show that survival and defense strategies, used by microorganisms to adapt to an environment, such as biofilm formation and quorum sensing, are less developed in the early stages of an infection. This would make antibiotic therapy potentially more effective in the early infection stage [69]. Unfortunately, there is so far no direct evidence of this hypothesis in bronchiectasis [94,95].

Treatment regimens aimed at eradication vary, but some evidence suggests that a treatment protocol, which includes nebulized antibiotics allows for greater clearance and clinical benefits than intravenous treatment alone [96,97,98,99]. Two studies examined whether eradication treatment in adults with bronchiectasis improved clinical outcomes relative to the patient’s underlying health status [96,97]. The pooled analysis provides some evidence of the potential benefits of *P. aeruginosa* eradication in terms of negative sputum samples, frequency of subsequent exacerbations, and QoL, but the evidence is indirect and considered to be of low quality. Notably, in 2012, White et al. conducted a retrospective observational study, which focused on different eradication treatment regimens: 12 patients treated with i.v. antibiotics, 13 patients with i.v. antibiotics (different combinations of antibiotics) followed by oral ciprofloxacin, and five patients treated with ciprofloxacin alone. Twenty-five patients in all groups received 3 months of nebulized colistin. The study reported that in 80% of enrolled patients *Pseudomonas aeruginosa* was initially eradicated, but only 54% of all patients remained free of *P. aeruginosa* at follow-up. Furthermore, the study demonstrated a reduced exacerbation rate from 3.93 to 2.09 during the year after the eradication treatment. Finally, two-thirds of patients reported clinical improvement even in the absence of changes in lung function. As stated by these authors, intravenous antibiotic treatment with inhaled antibiotics, potentially in combination with oral ciprofloxacin, may be considered, but oral ciprofloxacin coupled with nebulized colistin might also be an option [96].

In 2015, Orriols et al. randomized patients, after isolation of *P. aeruginosa*, to receive 300 mg nebulized tobramycin twice daily or placebo for 3 months and i.v. 14 days treatment with ceftazidime and tobramycin. Enrolled patients were then followed up for 12 consecutive months. The authors found that, at the end of the follow-up, 54.5% of patients were free of *P. aeruginosa* in the tobramycin group and 29.4% in the placebo group. Furthermore, tobramycin treatment was associated with a reduction in the number of exacerbations, hospital admissions and days of hospitalization. Finally, this study documented that tobramycin treatment has a favourable clinical impact. However, episodes of bronchospasm associated with nebulization of the drug should not be overlooked [97]. Despite very low-quality evidence, the European Respiratory Society guidelines [69] recommend eradication treatment in bronchiectasis patients with a newly isolated *P. aeruginosa* strain. Instead, the same guidelines advice against eradication treatment for adults with new isolates other than *P. aeruginosa*, because of lacking evidence [69]. The SEPAR guidelines recommend to consider Methicillin-resistant *Staphylococcus aureus* (MRSA) eradication based on knowledge obtained in CF patient treatment and expert opinions, even though evidences are lacking [71]. 

## 4. Antibiotic Resistance in Bronchiectasis

Lungs’ microbial ecology, antibiotic type, and administration consistency can strongly affect antibiotic resistance risks in patients with bronchiectasis. Past treatment errors caused a globally known health threat, known as MDR, with negative clinical practice impacts if inadequately identified and treated. In the following a list of pathogens with their MDR features: (1) *Enterobacteriaceae*: extended-spectrum-β-lactamase (ESBL) producing, resistance to most β-lactam antibiotics, such as cephalosporins, penicillin, and aztreonam; (2) *P. aeruginosa*: resistance against at least one agent in three or more antimicrobial categories, and; (3) *S. aureus*: oxacillin resistance with a MIC of ≥4 mcg/mL [100]. Chronic colonization with pathogens occurs more frequently in bronchiectasis patients causing recurrent exacerbations and infections for which patients take several broad-spectrum antibiotics. Those treatment cycles facilitate the development of MDR pathogens, which are frequently diagnosed in bronchiectasis patients, because of their specific treatment needs, restricted diversity of antibiotics and the fact that ESBL-producing *Enterobacteriaceae* and *P. aeruginosa* are likely MDR pathogens, which are very aggressive in patients with respiratory disorders necessitating specific antibiotic treatment, usually not recommended in guidelines [101].

During episodes of exacerbation, MDR pathogens are frequently isolated from patients with bronchiectasis, particularly if hospitalized. Pseudomonas, MRSA and ESBL + *Enterobacteriaceae* are the most frequently encountered MDR bacteria. Several risk factors are independently associated with the isolation of MDR bacteria—the most frequent being previous MDR isolation, hospitalization in the previous year and chronic kidney disease. Studies focusing on pneumonia recognize chronic kidney disease as a risk factor for MDR bacteria. Another widely recognized independent risk factor for MDR, particularly MRSA and *Enterobacteriacea*, is previous hospitalization for exacerbation, especially if there has been excessive or inadequate use of 3rd/4th generation cephalosporins or broad spectrum penicillins in this condition. MDRs are more frequently hospital related and not acquired in the community; their incidence and spectrum corresponds to the treatment attitude towards these challenging pathogens considering preceding antibiotic therapy and patient characteristics [42]. Furthermore, it is known that elderly patients, especially those suffering from multiple comorbidities, are at greater risk of exacerbations from MDR bacteria, indicating possible associations between risk of MDR infection and more debilitating diseases, previous use of inhaled antibiotics, and long-term oxygen therapy [101].

These risk factors are common in patients over 65 years with bronchiectasis, making the impact of MDR bacteria significant, especially during flare-ups. Hence, performing sputum cultures during the flare-up phases of bronchiectasis is important to optimize the choice of antibiotic therapy and to avoid excessive use of broad-spectrum antibiotics. It is essential to modify empirical therapy based on microbiological results and antibiotic sensitivity tests as soon as the results of the sputum culture are available. Nevertheless, empirical antibiotic therapy should be based on preceding microbiological findings until new results are available.

The use of broad-spectrum antibiotics against MDR pathogens must be based on the presence or absence of risk factors. In fact, the use of these antibiotics is indicated if two or more risk factors are present. Whether or not a broad-spectrum antibiotic is confirmed will depend on a microbiological work-up. This strategy allows minimizing broad-spectrum coverage for MDR in episodes of exacerbation for which such antibiotics are not needed and reducing the future appearance of resistant microorganisms. In general, MDR pathogens are related to longer hospital stays, increased need for antibiotics and utilization of healthcare resources, and may adversely affect patient outcomes [101].

In the treatment of chronic infection, long-term use of macrolides may lead to an increased risk of antimicrobial resistance (Table 4). In fact, the BAT study showed that after 12 months of treatment with azithromycin, patients had 88% macrolide resistance rate compared to 26% observed in the placebo group. Interestingly, over 80% of the overall pathogens (*H. influenzae*, *S. pneumoniae*, *M. catarrhalis*, and *H. parainfluenzae*), considered to be chronically present in the airways of recruited patients, were potentially sensitive to azithromycin, while only 11% of all enrolled patients were infected with *P. aeruginosa*, a bacterium naturally resistant to macrolides [80]. Hence, attention needs to be paid to chronic treatment with macrolides, which should be reserved only for patients with bronchiectasis with more than three exacerbations per year. The results of the BLESS study also show that treatment with macrolide (erythromycin, 400 mg, administered twice daily consecutively over 12 months) increases the percentage of macrolide-resistant oropharyngeal streptococci [79]. The clinical relevance of the increased risk of antibiotic resistance resulting from the long-term use of macrolides must be further investigated considering also characteristics and microbiological data of patients with bronchiectasis. To date, the major concern about the long-term use of macrolides is the possible impact on antibiotic resistance of NTM. Indeed, newer macrolides such as azithromycin and clarithromycin are the first-line therapy for NTM infection due to their direct antimicobacterial action [102]. Resistance to macrolides by ubiquitous microorganisms such as staphylococci, streptococci and *Haemophilus* can also be found in patients with community-acquired pneumonia [103]. Furthermore, the intake of antibiotics can change the composition of the microbiome. Indeed, Wang et al. [104] demonstrated a higher prevalence of gram-negative organisms in patients who have had a COPD exacerbation, with a subsequent shift towards a predominance of gram-positive organisms after antibiotic administration. Finally, long-term administration of antibiotics could potentially induce substantial changes in the respiratory microbiome [104]. This was further confirmed by Rogers et al. [105]. Indeed, in a post hoc analysis of the BLESS study, the authors analyzed changes in microbiome composition after 48 weeks of erythromycin administration. The authors demonstrated that long-term treatment with erythromycin increased the positivity rate of *P. aeruginosa* in patients initially colonized by microorganisms other than *P. aeruginosa*. In contrast, patients who were already colonized by *P. aeruginosa* did not have any changes in the respiratory microbiome. Hence, the choice of long-term treatment with erythromycin in patients with bronchiectasis not colonized by *P. aeruginosa* needs to be carefully considered [105].

Lower risks of MDR and systemic adverse reactions frequently favored nebulized antibiotics over systemic antibiotics in CF and NCFB patients (Table 5). 

The higher drug concentrations in the airways, which are reached by inhaled antibiotics, compared with systemic administration, and the minimal systemic absorption of the drug through the alveolar-capillary barrier support this therapeutic approach. Furthermore, the administration of inhaled antibiotics may help to contain the increase of antimicrobial resistance in bronchiectasis patients with chronic infections.

The AIR-BX1 and AIR-BX2 trials demonstrated that tobramycin-resistant *P. aeruginosa* strains developed in 11% of tobramycin-treated patients compared with 3% of placebo-treated patients (*p* = 0.36) [91]. However, in a double blind, placebo-controlled crossover study focusing on the safety and efficacy of tobramycin, Drobnic et al. found no differences in antibiotic resistance [106]. More recently, the ORBIT-3 and ORBIT-4 studies, which focused on the safety and efficacy of liposomal ciprofloxacin, did not document any significant reduction in antibiotic activity over 48 weeks in over 1000 patients enrolled worldwide [89]. The RESPIRE 1 and RESPIRE 2 studies, which involved the use of dry powder ciprofloxacin, documented a significant increase in the MIC of the pathogens in both treatment arms (14 days on/off and 28 days on/off). In the RESPIRE 1 trial, 24.5% of patients had a pathogen with an elevated MIC for ciprofloxacin at baseline. The number of patients with elevated MIC pathogens was 20.4%, 26.2%, and 12.3% for ciprofloxacin DPI 14 days on/off, ciprofloxacin DPI 28 days on/off, and pooled placebo, respectively. In the RESPIRE 2 trial, 18.8% of patients had a pathogen with an elevated minimal MIC for ciprofloxacin at baseline. The number of patients with one isolate from sputum with an elevated MIC from pre-treatment at any time point during the study was 21.0% for ciprofloxacin DPI 14 days on/off, 16.5% for ciprofloxacin DPI 28 days on/off, and 9.8% for pooled placebo [87,88]. The AIR-BX 1 and AIR-BX 2 studies, focusing on the use of nebulized aztreonam, described an increase in MIC in 15–23% of enrolled patients [91]. Finally, Haworth et al. found no significant increase in resistant *P. aeruginosa* strains in a study on inhaled colistin [107].

To date, studies exploring the use of gentamicin, colistin, and tobramycin have not demonstrated a significant emergence of antimicrobial resistant isolates in sputum. In addition, any increase in MIC was transient with return to baseline after discontinuation of treatment. Finally, few data support a possible causal link between the reduction of the bacterial load or the apparent eradication of the dominant pathogen and the establishment of treatment-emergent pathogens. Reasons that may explain the absence of resistance development are diverse and multifactorial. First, it is likely that traditional MIC, determined in combination with parenteral breakpoints, is not applicable to inhaled antibiotics, because there would be significantly increased concentrations in the sputum to allow for safe parenteral administration without toxic effects. Further research is needed to define adjusted airway MICs, which more directly reflect concentrations achieved by inhaled antibiotics. Furthermore, a transient increase in MIC, especially if clinically insignificant, could be abundantly offset by the benefits of reducing exacerbations and therefore by exposure to systemic antimicrobial therapies.

Indeed, unlike the significant antimicrobial resistance observed with macrolide therapy, the administration of inhaled antibiotics resulted in only a modest increase in resistant strains in a variable proportion of patients based on study drug and formulation (solution for inhalation, dry or liposomal powder).

Ultimately, multiple factors must be considered when choosing between oral and inhaled antibiotics: in addition to the risk of antimicrobial resistance, patient characteristics, comorbidities, concomitant drugs, and expected benefits in terms of exacerbations and QoL must be considered.

## 5. Five-Year View

To date, the management of patients with acute exacerbation of bronchiectasis, in particular when sustained by MDR microorganisms, represents one of the most difficult treatment challenges in respiratory diseases.

It is well known that the timely administration of appropriate, pathogen-directed therapies is crucial. Because results of culture and antimicrobial susceptibility testing can take 48–72 h or longer, physicians rely currently on clinical and epidemiological factors including previous microbiological isolates, if available, to support the choice of empiric antibiotic therapy.

Currently, clinical microbiology laboratories usually adopt automated susceptibility testing systems, which require at least 48 h to yield a result. Furthermore, susceptibility testing of some antibiotics may affect the performance of such tests.

Therefore, research and development of rapid molecular tests able to identify both pathogens and genetic determinants of antimicrobial resistance are crucial to contain the growing phenomenon of antibiotic resistance. The basis for most molecular assays includes polymerase chain reaction (PCR, which amplifies DNA) or reverse-transcription PCR (RT-PCR) and nucleic-acid-sequence-based amplification. The bacterial DNA of 16S ribosomal RNA (rRNA) genes or 16S–23S rRNA gene spacer regions are the common target of most molecular assays [108].

To date, multiple rapid molecular tests could be increasingly used to manage patients suffering from bronchiectasis, particularly during acute exacerbations. Among these tests, the authors include Cepheid’s GeneXpert system (Sunnyvale, United States), AccuProbe (Gen-Probe, San Diego, United States), BD GeneOhm StaphSR and BD GeneOhm MRSA assays (Eysins, Vaud. Switzerland), ResPlex and StaphPlex panels (Germantown, United States), Molecular Beacons, FilmArray System (Marcy l'Etoile, France), Microarray Technologies Detecting β-Lactamases, and PCR Followed by ElectroSpray Ionization MS (PCR/ESI-MS).

## 6. Conclusions

Conclusively, more research is required in the field of bronchiectasis to identify worthwhile and valuable actions tackling antimicrobial resistance globally. Without doubt, continuous follow-up of pathogenic microorganisms is mandatory to prevent chronic lung infections resulting in long-term antibiotic treatment, which can only be avoided with early detection and adequate antibiotic therapy. In consideration of growing bacterial resistance to antibiotics, new drugs are necessary. In early-phase clinical trials, enrolling patients with exacerbations, several novel anti-pseudomonal drugs are presently studied, among them a novel protegrin-based (an antimicrobial peptide) anti-pseudomonal drug [109,110]. Moreover, in the future, the development of affordable microbiological rapid diagnostic and susceptibility tests could improve the management of infections in bronchiectasis to minimize the overuse of broad-spectrum antibiotics, which contribute to the spread of antimicrobial resistance. The efficacy of other strategies for bronchiectasis patients, such as patient segregation in case of MDR pathogens, should be investigated in terms of cross-infection risk in healthcare settings. Finally, the role of respiratory vaccines (e.g., influenza viruses, *S. pneumoniae* and, from the year 2020, also severe acute respiratory syndrome coronavirus 2-SARS-CoV-2), to reduce the risk of antibiotic overuse and resistance in bronchiectasis, should be further investigated.

## Figures and Tables

**Figure 1 antibiotics-10-00326-f001:**
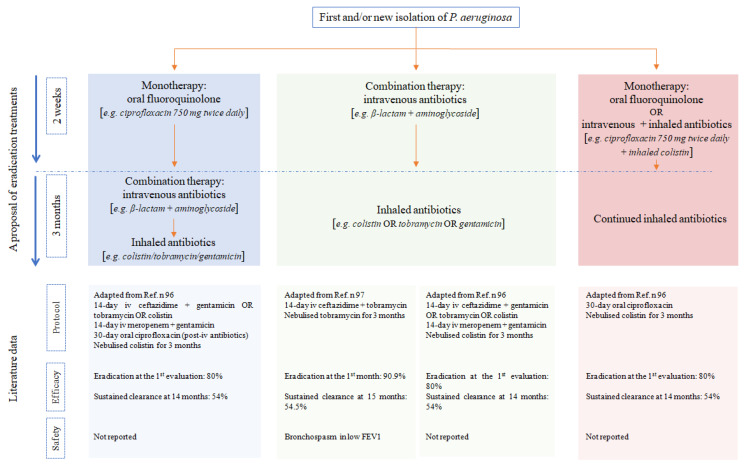
A proposal for eradication treatment pathways for isolation of *P. aeruginosa.*

**Table 1 antibiotics-10-00326-t001:** Bacteriology of bronchiectasis.

	Nicotra et al. (1995) (123 pts) Ref. n33	Pasteur et al. (2000) (150 pts) Ref. n34	Aksamit et al. (2017) (1406 pts) Ref. n35	Dimakou et al. (2016) (205 pts) Ref. n37	Martinez-García et al (2020) * (849 pts) Ref. n40	McDonnell et al. (2015) (155 pts) Ref. n42	King et al. (2007) (89 pts) Ref. n44	Cabello et al. (1997) (17 pts) Ref. n45	Venning et al. (2017) * (65 pts) Ref. n63
*Haemophilus influenza*	37 [30]	**52 [35]**	116 [8]	26 [13]	**[14]**	**89 [57]**	**42 [47]**	**10 [42]**	[15]
*Streptococcus pneumoniae*	13 [11]	20 [13]	49 [3]	17 [8]	[5]	51 [33]	6 [7]	0 [0]	N/R
*Staphylococcus aureus*	9 [7]	21 [14]	170 [12]	N/R	[4]	35 [23]	3 [4]	**4 [17]**	[3]
*Pseudomonas aeruginosa*	**38 [31]**	**46 [31]**	**470 [33]**	**88 [43]**	**[26]**	**76 [49]**	11 [12]	1 [4]	**[32]**
Mycobacteria	**49 [40]**	0 [0]	**657 [50**]	2 [1]	[2]	5 [3]	2 [2]	N/R	[<3]
No organism	N/R	34 [23]	93 [7]	**78 [38]**	N/R	N/R	**19 [21]**	N/R	**[17]**

The most frequent microbiological findings are in bold. N/R: not reported. *: Studies in which the Authors report microbiological data only as percentages of the total sample.

**Table 2 antibiotics-10-00326-t002:** Recommended antibiotic treatment according to the most common microorganisms found in exacerbated bronchiectasis.

Microrganism		Recommended First-Line Treatment (14 Days)		Recommended Second-Line Treatment (14 Days)
*Hemophilus influenzae*—beta lactamase positive		Amoxicillin/clavulanic acid 625 mg 1 tablet three times a day *~		Doxycycline 100 mg twice a day *~
				Ciprofloxacin 500 mg or 750 mg twice a day *~†
		Amoxicillin/clavulanic acid 825 mg 1 tablet three times a day †		Ceftriaxone 2 g once a day (intravenous) *~
*Moraxella catarrhalis*		Amoxicillin/clavulanic acid 625 mg 1 tablet three times a day *		Clarithromycin 500 mg twice a day *
				Doxycycline 100 mg twice a day *
				Ciprofloxacin 500 mg or 750 mg twice a day *
*Streptococcus pneumoniae*		Amoxicillin 500 mg three times a day *		Doxycycline 100 mg twice a day *
*Staphylococcus aureus* (MSSA)		Flucloxacillin 500 mg four times a day *		Clarithromycin 500 mg twice a day *
				Doxycycline 100 mg twice a day *
				Amoxicillin/clavulanic acid 625 mg 1 tablet three times a day *
*Staphylococcus aureus* (MRSA)	Oral route	Doxycycline 100 mg twice a day *		Third-line linezolid 600 mg twice a day *
		Rifampicin (<50 kg) 450 mg once a day *		
		Rifampicin (>50 kg) 600 mg once a day *		
		Trimethoprim 200 mg twice a day *		
	Intravenous route	Vancomycin 1 g twice a day *		Linezolid 600 mg twice a day *
		Teicoplanin 400 mg once a day *		
*Pseudomonas aeruginosa*		Oral ciprofloxacin 500 mg twice a day *	Monotherapy	intravenous ceftazidime 2 g three times a day *
		Oral ciprofloxacin 750 mg twice a day in more severe infections *†		piperacillin with tazobactam 4.5 g three times a day *
				aztreonam 2 g three times a day *
				meropenem 2 g three times a day *
			Dual therapy	Previous drugs combined with gentamicin or tobramycin or colistin 2 mU three times a day (under 60 kg, 50 000–75 000 U/kg daily in 3 divided doses) *

†: Spanish Respiratory Medicine Society Guidelines (SEPAR) (Ref. n 71); *: British Thoracic Society Guidelines (BTS) (Ref. n 1); ~: National Institute for Health and Care Excellence guidelines (NICE) (Ref. n 72).

**Table 3 antibiotics-10-00326-t003:** Management of chronic infection: pharmacological treatment.

If ≥ 3 Exarcerbations/Year [I Step]		If ≥ 3 Exarcerbations/Year Despite I Step Treatment [II Step]	If ≥ 5 Exacerbations/Year Despite II Step Treatment [III Step]
*Pseudomonas aeuriginosa*	long term inhaled anti-pseudomonal OR long term macrolide *		
Other potentially pathogenic microorganisms	long term macrolides * OR long term oral targeted antibiotic OR long term inhaled targeted antibiotic	long term macrolide * AND long term inhaled antibiotic	Regular intravenous antibioticevery 2–3 months
No pathogen	long term macrolides *		

*****: Azithromycin is administered in a dose range from 250 mg or 500 mg three times per week up 250 mg daily. Erythromycin is administered 400 mg twice daily. Adapted from Ref. n 1.

**Table 4 antibiotics-10-00326-t004:** Trials with long-term macrolides in bronchiectasis.

Trial (Ref n)	Inclusion Criteria	Intervention	Duration	Primary end Point	Main Results	Antibiotics Resistance (MDR)
**BLESS** (79)	≥ pulmonary exarcerbations requiring supplemental systemic antibiotic therapy in the preceding 12 months and daily sputum production	Erythromycin 400mg every 12 h versus placebo	12 months	The mean rate of PDPEs per patient per year, analyzed by intention to treat	Significant reduction of PDPEs in the erythromycin group	Median % of macrolide resistant oropharyngeal streptococci: 25.6
	Age 20–85 years					No difference for the emergence of new sputum pathogens
**BAT** (80)	≥3 LRTIs treated with oral or i.v. antibiotics and ≥1 sputum culture yielding one or more bacterial respiratory pathogens in the previous year	Azythromycin 250 mg daily versus placebo	12 months	N° of infectious exacerbations during the 52-week treatment period.	Zero exacerbations in the azithromycin group	% of macrolid resistance in the azithromycin group: 88% versus 26% in placebo group
	≥18 years					
**EMBRACE** (81)	≥1 pulmonary exarcerbation requiring antibiotic treatment in the past year ≥18 years	Azythromycin 500 mg days week	6 months of treatment, followed up for another 6 months	Rate of event-based exacerbations in the first 6 months	62% relative reduction with azithromycin in the 6-month treatment period. 42% relative reduction in the 12-month period.	Not routinely undertaken, but two (4%) patients in the azithromycin group developed macrolide-resistant Streptococcus pneumoniae at 6 months
				FEV1 before bronchodilation	No significant changes	
				SGRQ total score at the end of the treatment period	No significant changes	

**Table 5 antibiotics-10-00326-t005:** Trials with long-term inhaled antibiotics in bronchiectasis.

1st Author or Trial (Ref n)	Inclusion Criteria	Sputum Bacteriology	Intervention	Duration	Primary End Point	Main Results	Antibiotics Resistance (MDR)
**RESPIRE 1 and 2** (87, 88)	≥2 exacerbations in the previous 12 months	*P. aeruginosa*, *H. influenzae*, *M. catarrhalis*, *S. aureus*, *S. pneumoniae*, *S. maltophilia*, *B. cepacia*	Ciprofloxacin DPI 32.5 mg every 12 h	1 year, 14 days on/off (12 active cycles) or 28 days on/off (six active cycles)	(1) time to first exacerbation AND (2) frequency of exacerbations	Ciprofloxacin DPI 14 days on/off delayed time to 1st exacerbation AND significantly reduced frequency of exacerbations by 39%	% of patients with ≥1 isolate from sputum with an elevated MIC at any time-point: **54.0%** for ciprofloxacin DPI 14 days on/off and **53.9%** for ciprofloxacin DPI 28 days on/off versus **36.2%** for placebo
**ORBIT-3 and -4** (89)	≥ pulmonary exacerbations treated with antibiotics in the preceding 12 months AND history of chronic *P aeruginosa* lung infection	*P. aeruginosa*	ARD-3150 (liposome encapsulated ciprofloxacin 135 mg and free ciprofloxacin 54 mg)	1 year, on/off regimen (six active cycles)	Occurance of pulmonary exacerbations	Reduction of pulmonary exacerbations of all severity in ORBIT-4, but not in ORBIT-3, compared with placebo	**32%** of patients treated with ARD-3150 and **18%** of patients treated with placebo had a *P. aeruginosa* isolate for which the ciprofloxacin MIC had increased by > 2 times
**Murray** (90)	Chronically infected sputum AND ≥2 exacerbations in the past year AND ability to tolerate nebulized gentamicin AND FEV1 > 30% predicted AND not currently receiving long-term antibiotics	Any PPM	Gentamicin 80 mg every 12 h	1 year, continuous regimen	≥1 log unit reduction in sputum bacterial density	Bacterial density significantly reduced in the gentamicin group. At follow-up: bacterial density was similar in both groups	No difference for the emergence of gentamicin indeterminately resistant or resistant strains
**AIR-BX 1 and 2** (91)	History of positive sputum or bronchoscopic culture for target Gram-negative organism or treatment of exacerbation AND chronic sputum production AND FEV1 ≥ 20% predicted	*P. aeruginosa*, *Achromobacter*, *Burkholderia*, *Citrobacter*, *Enterobacter*, *Escherichia*, *Klebsiella*, *Moraxella*, *Proteus*, *Serratia*, *Stenotrophomonas*	Aztreonam solution 75 mg every 8 h	4 months, 28 days on/off (two active cycles)	Δ in QOL-B-RSS (baseline to week 4; high scores represent few symptoms)	QOL-B-RSS numerically increased in all groups in both studies at weeks 4 and 12. No significant differences	Increases of ≥4 fold in the MIC of aztreonam: (A) in AIR-BX1: **15%** of AZLI-treated patients versus **6%** of placebo after 4 weeks; **35%** versus **11%** after 12 weeks; and **23%** versus **14%** of placebo after 4 weeks off -treatment. (B) AIR-BX2: **23%** of AZLI-treated patients versus **7%** after 4 weeks; **34%**, versus **11%** after 12 weeks; and **20%** versus **6%** after 4 weeks off -treatment
**Orriols** (97)	Recruitment after the 1st isolation of *P. aeruginosa* in sputum	*P. aeruginosa*	Nebulized tobramycin 300 mg every 12 h + i.v. ceftazidime	14 days during the first 4 weeks, then randomization and treatment for 3 months	Bacterial eradication in sputum	% of patients free of *P. aeruginosa*: (A) in the 1st month: 90.9% in tobramycin group versus 76.5% in placebo. (B) At the end of study: 54.5% in tobramycin group versus 29.4% in placebo	No tobramycin-resistant *P. aeruginosa*
**Drobnic** (106)	≥3 positive sputum cultures for tobramycin-sensitive *P. aeruginosa* during 6 months prior to the study	*P. aeruginosa*	Tobramycin 300 mg every 12 h	6 months	N° of exacerbations AND days of hospital admissions	No significant differences in the frequency of pulmonary exacerbations. Days of hospital admission significantly fewer in the tobramycin period	2 months after ending the study, all patients remained colonized by tobramycin-susceptible PA (MIC < 8 µg/mL)
**Haworth** (107)	≥2 positive respiratory tract cultures for *P. aeruginosa* in the preceding 12 months AND within 21 days of completing a course of antipseudomonal antibiotics for the treatment of an exacerbation	*P. aeruginosa*	Colistin 1 million IU every 12 h	6 months, continuous regimen	Time to exacerbation	The median time to exacerbation was 165 days in the colistin group versus 111 days in the placebo group	No colistin-resistant strains of *P. aeruginosa*

**PPM**: potentially pathogenic micro-organism; **DPI**: dry powder inhalation; **QOL-B RSS**: Quality of Life-Bronchiectasis respiratory symptoms domain score; **MDR**: multidrug-resistant.

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
