# Peer review of "Antimicrobial Resistance in Common Respiratory Pathogens of Chronic Bronchiectasis Patients: A Literature Review"

_antibiotics, 2021, doi:10.3390/antibiotics10030326_

Round 1

Reviewer 1 Report

While this manuscript discussed important topic especially in a time of improving antimicrobial stewardship, there is lack of novelty and have been previously discussed before.

English native editing is required with better scientific language. Multiple grammar mistakes and incorrect word usage. The logical sequences  need major improvement to improve readability.

Introduction lacks a more comprehensive review on pathophysiology and risk factors to include acquired and inherited immunodeficiencies. The logical sequence of introduction needs to be reviewed and improved.

A future perspective or personal insight before conclusion will add authenticity to the paper. This should include more on rapid testing and the role of GeneXpert or mutation platform to guide treatment.

For chronic infection section, a table or figure to illustrate recommended treatment would be consistent with section 3.1 and 3.3 especially if the manuscript is used as a quick guide or reference.

Author Response

March, 13th 2021

To the Editor and Reviewers

Antibiotics

We would like to thank the Editor and Reviewers who encouraged a complete revision of the manuscript.

Please find enclosed the Revision vers. 1 of the Review entitled “Antimicrobial resistance in common respiratory pathogens of chronic bronchiectasis patients: A literature review” by Riccardo Inchingolo, Chiara Pierandrei, Giuliano Montemurro, Andrea Smargiassi, Franziska Michaela Lohmeyer and Angela Rizzi.

[Antibiotics] Manuscript ID: antibiotics-1122144 - Major Revisions

Author's Reply to the Review Report (Reviewer 1)

Comments and Suggestions for Authors

GENERAL COMMENTS

  1. While this manuscript discussed important topic especially in a time of improving antimicrobial stewardship, there is lack of novelty and have been previously discussed before. English native editing is required with better scientific language. Multiple grammar mistakes and incorrect word usage. The logical sequences need major improvement to improve readability.

We thank the Reviewer for the comment. The manuscript was revised by a member of American Medical Writers Association.

  1. Introduction lacks a more comprehensive review on pathophysiology and risk factors to include acquired and inherited immunodeficiencies. The logical sequence of introduction needs to be reviewed and improved.

We thank the Reviewer for the comment. The Introduction section has been extensively modified and expanded according to Reviewer’s indications. The logical sequence has also been changed.

  1. A future perspective or personal insight before conclusion will add authenticity to the paper. This should include more on rapid testing and the role of GeneXpert or mutation platform to guide treatment.

We thank the Reviewer for the comment. We added the section “Five-year view” according to Reviewer’s comments.

  1. For chronic infection section, a table or figure to illustrate recommended treatment would be consistent with section 3.1 and 3.3 especially if the manuscript is used as a quick guide or reference.

We thank the Reviewer for the comment. We added Table 3 in order to describe pharmacological treatments in the management of chronic infection. Furthermore, we added other two Tables (4 and 5) in order to both detail pharmacological treatments in chronic infection (macrolides and inhaled drugs) and summarize the studies about antibiotics resistance (MDR) as requested by the other Reviewer.

With the best regards,

Riccardo Inchingolo, Chiara Pierandrei, Giuliano Montemurro, Andrea Smargiassi, Franziska Michaela Lohmeyer and Angela Rizzi

Corresponding Author:

Riccardo Inchingolo, MD, PhD

UOC Pneumologia, Fondazione Policlinico Universitario A. Gemelli IRCCS. Largo A. Gemelli, 8 – 00168 – Rome, Italy.

[email protected]

Corresponding Author will receive all editorial communications

The authors declare that the manuscript, or specified parts of it, have not been and will not be submitted elsewhere for publication.

Reviewer 2 Report

The review article is comprehensive and readable, it is worthy to be published in “Antibitoics”, some suggestions were proposed.

  1. In Table 2, please cite the references of guidelines, different guidelines may have different antibiotics recommendation in different studies and regions, it also has local resistant issues.
  2. In Fig 1. The title is “Eradication treatment pathways for the treatment of adult bronchiectasis”, However, the flow chart only enrolls “isolation of P. aeruginosa”, please modify it, it would be better to include the successful rate and complications to help physicians to make drugs decision. In addition, the figure originated from ERS guideline 2017, but the original study should be cited, it will make the recommendation reasonable.
  3. Can authors summarize the studies about Antibiotics resistance (MDR) in the bronchiectasis, such as BLESS study, AIR-BX1, AIR-BX2 trials, ORBIT-3 and ORBIT-4 studies, RESPIRE 1 and RESPIRE 2 studies and so on. The issue is critical, and I believed the summary make the manuscript more readable.

Author Response

March, 13th 2021

To the Editor and Reviewers

Antibiotics

We would like to thank the Editor and Reviewers who encouraged a complete revision of the manuscript.

Please find enclosed the Revision vers. 1 of the Review entitled “Antimicrobial resistance in common respiratory pathogens of chronic bronchiectasis patients: A literature review” by Riccardo Inchingolo, Chiara Pierandrei, Giuliano Montemurro, Andrea Smargiassi, Franziska Michaela Lohmeyer and Angela Rizzi.

[Antibiotics] Manuscript ID: antibiotics-1122144 - Major Revisions

Author's Reply to the Review Report (Reviewer 2)

Comments and Suggestions for Authors

The review article is comprehensive and readable, it is worthy to be published in “Antibitoics”, some suggestions were proposed.

  1. In Table 2, please cite the references of guidelines, different guidelines may have different antibiotics recommendation in different studies and regions, it also has local resistant issues.

We thank the Reviewer for the comment. The Table 2 was modified accordingly.

  1. In Fig 1. The title is “Eradication treatment pathways for the treatment of adult bronchiectasis”, However, the flow chart only enrolls “isolation of P. aeruginosa”, please modify it, it would be better to include the successful rate and complications to help physicians to make drugs decision. In addition, the figure originated from ERS guideline 2017, but the original study should be cited, it will make the recommendation reasonable.

We thank the Reviewer for the comment. The title of the Figure 1 was modified accordingly. Furthermore, the figure was enriched including both successful rate and complications derived from published studies and reporting correspondent references.

  1. Can authors summarize the studies about Antibiotics resistance (MDR) in the bronchiectasis, such as BLESS study, AIR-BX1, AIR-BX2 trials, ORBIT-3 and ORBIT-4 studies, RESPIRE 1 and RESPIRE 2 studies and so on. The issue is critical, and I believed the summary make the manuscript more readable.

We thank the Reviewer for the comment. We added two Tables (4 and 5) in order to both summarize the studies about antibiotics resistance (MDR) and detail pharmacological treatments in chronic infection (macrolides and inhaled drugs), as requested by the other Reviewer.

With the best regards,

Riccardo Inchingolo, Chiara Pierandrei, Giuliano Montemurro, Andrea Smargiassi, Franziska Michaela Lohmeyer and Angela Rizzi

Corresponding Author:

Riccardo Inchingolo, MD, PhD

UOC Pneumologia, Fondazione Policlinico Universitario A. Gemelli IRCCS. Largo A. Gemelli, 8 – 00168 – Rome, Italy.

[email protected]

Corresponding Author will receive all editorial communications

The authors declare that the manuscript, or specified parts of it, have not been and will not be submitted elsewhere for publication.